# HBV-RNA, Quantitative HBsAg, Levels of HBV in Peripheral Lymphocytes and HBV Mutation Profiles in Chronic Hepatitis B

**DOI:** 10.3390/v14030584

**Published:** 2022-03-11

**Authors:** Yael Gozlan, Daniella Aaron, Yana Davidov, Maria Likhter, Gil Ben Yakov, Oranit Cohen-Ezra, Orit Picard, Oran Erster, Ella Mendelson, Ziv Ben-Ari, Fadi Abu Baker, Orna Mor

**Affiliations:** 1Central Virology Laboratory, Ministry of Health, Chaim Sheba Medical Center, Ramat Gan 52621, Israel; daniellame36@gmail.com (D.A.); oran.erster@sheba.health.gov.il (O.E.); ella.mendelson@sheba.health.gov.il (E.M.); 2Sackler Faculty of Medicine, Tel-Aviv University, Tel Aviv 69978, Israel; ziv.benari@sheba.health.gov.il; 3The Center for Liver Diseases, Sheba Medical Center, Ramat Gan 52621, Israel; yana.davidov@sheba.health.gov.il (Y.D.); mariya.likhter@sheba.health.gov.il (M.L.); gil.benyakov@sheba.health.gov.il (G.B.Y.); oranit.cohenezra@sheba.health.gov.il (O.C.-E.); 4Gastroenterology Laboratory, Sheba Medical Center, Ramat Gan 52621, Israel; orit.picard@sheba.health.gov.il; 5Hillel Yaffe Medical Center, The Gastroenterology Institute, Hadera 38100, Israel; fa_fd@hotmail.com

**Keywords:** hepatitis B, HBV-RNA, HBV biomarkers, HBeAg negative

## Abstract

A comprehensive characterization of chronic HBV (CHB) patients is required to guide therapeutic decisions. The cumulative impact of classical and novel biomarkers on the clinical categorization of these patients has not been rigorously assessed. We determined plasma HBV-RNA and HBsAg levels, HBV in peripheral lymphocytes (PBMCs) and HBV mutation profiles in CHB patients. Patient demographics (*n* = 139) and classical HBV biomarkers were determined during a clinical routine. HBV-RNA in plasma and HBV-DNA in PBMCs were determined by RT-PCR. HBsAg levels were determined using Architect. In samples with HBV-DNA viral load >1000 IU/mL, genotype mutations in precore (PC), basal core promoter (BCP), HBsAg and Pol regions were determined by sequencing. Most patients (*n* = 126) were HBeAg-negative (HBeAgNeg) with significantly lower levels of HBV-RNA, HBV-DNA and HBsAg compared to HBeAg-positive (HBeAgPos) patients (*p* < 0.05). HBV genotype D prevailed (61/68), and >95% had BCP/PC mutations. Escape mutations were identified in 22.6% (13/63). HBeAgNeg patients with low levels of HBsAg (log IU ≤ 3) were older and were characterized by undetectable plasma HBV-DNA and undetectable HBV-RNA but not undetectable HBV-DNA in PBMCs compared to those with high HBsAg levels. In >50% of the studied HBeAgNeg patients (66/126), the quantitation of HBsAg and HBV-RNA may impact clinical decisions. In conclusion, the combined assessment of classical and novel serum biomarkers, especially in HBeAgNeg patients, which is the largest group of CHB patients in many regions, may assist in clinical decisions. Prospective studies are required to determine the real-time additive clinical advantage of these biomarkers.

## 1. Introduction

Chronic Hepatitis B virus (CHB) infection, which may progress to advanced fibrosis, cirrhosis, liver failure, and hepatocellular carcinoma (HCC), remains a global health problem [1,2]. Antiviral therapy, which targets the viral reverse transcriptase and blocks viral DNA synthesis, cannot directly eradicate the viral hepatic covalently closed circular DNA (cccDNA) and does not provide definitive resolution of the disease [3].

As the clinical course of CHB is variable, several classical non-invasive serum biomarkers, which reflect the status of HBV in the liver, have been introduced. These include plasma HBV-DNA (which reflects viral proliferation), the viral secreted antigens HBsAg (an embedded viral protein that is related to the viral envelope) and HBeAg (a pre-core protein), liver enzymes (alanine aminotransferase, ALT, and aspartate transaminase, AST, which are elevated in chronic HBV hepatitis patients), in combination with a non-invasive assessment of liver fibrosis [4]. According to these markers, and based on the two main characteristics of chronicity of viral hepatitis—infection versus hepatitis—CHB is classified into four main phases: HBeAg-positive chronic infection, HBeAg-positive chronic hepatitis, HBeAg-negative chronic infection, and HBeAg-negative chronic hepatitis [5].

Aiming to identify patients who require antiviral treatment, those who need close surveillance only, or those selected patients who can stop the direct HBV treatment according to current guidelines [5], novel markers have been suggested. Included is the quantitation of HBsAg levels, a marker that can help distinguish between different CHB phases [6]. Low levels (<1000 IU/mL) are thought to be predictive of a subsequent functional cure [7]. Quantitation of HBV-RNA in plasma is another such marker, which is considered a direct marker for cccDNA transcriptional activity [8]. Levels of HBV-DNA in peripheral blood mononuclear cells (PBMCs) were also suggested as a potential marker for intrahepatic HBV levels [9], although there is limited evidence to support this assumption.

HBV is divided into 10 distinct genotypes, designated A–J. The risk of developing severe liver disease and the response to antiviral treatment are known to deviate between these genotypes [10]. For example, HBV genotypes C, D, and F are associated with a higher lifetime risk of cirrhosis and HCC compared to other genotypes [11]. Moreover, mutations in the viral genome have been shown to influence the natural course of the disease: studies indicate that mutations in preS/S1 (Pol region) are related to vaccine failure and immune escape (escape mutations), and those in polymerase (Pol region) lead to drug resistance to treatment with nucleos(t)ide analogues (NA) antivirals [12]. Mutations in the basal core promoter (BCP) and in the pre-core (PC) affect HBeAg production and are also associated with moderate-to-severe liver disease [13]. Defective HBeAg secretion was mainly related to A1762T, G1764A (BCP), G1896A, or G1899A (PC) mutations. PC mutations that affect the HBeAg initiation codon (positions 1814–1816) and the 1874 nonsense mutation have also been shown to lead to HBeAg sero-negativity [14,15]. A previous study in Israel, which was based on a cohort of 81 patients, found that the most prevalent genotype was genotype D, and that mutations in the precore were detectable in most patients with persistent hepatitis B virus (HBV) infection [16,17].

To date, both classical and new molecular and serological non-invasive biomarkers have not been rigorously assessed together in various cohorts and within different patient populations. Here, we determined levels of HBV-RNA and HBsAg in plasma, quantified HBV-DNA in lymphocytes, and analyzed HBV escape, resistance, and precore/core mutations; then, we combined the results of these parameters with data collected on classical biomarkers from a cohort of both CHB treatment-naïve and treated patients in Israel. In addition, the potential value of these markers was assessed in the aspect of personalized medicine.

## 2. Materials and Methods

### 2.1. Study Population

CHB patients who followed up in the liver units of the Sheba and Hillel Yaffe Medical Centers during the period December 2019–August 2021 were approached. In total, 139 CHB patients agreed to participate in this study. Blood was collected, and serum, plasma, or PBMCs (see below) were separated and stored at −20 °C. Levels of HBV-DNA and of liver enzymes (ALT, AST), as well as fibrosis stage (assessed by vibration-controlled transient elastography, Fibroscan) were all collected from electronic medical records (EMR) and determined in clinical routine. In cases where HBV-DNA levels were not available, they were assessed herein as previously described [18]. The study was approved by the local institute review board (number 632419), and written informed consent was obtained from all patients.

### 2.2. HBsAg Quantitation and Qualitative Determination of HBeAg, Anti-HBeAg and Anti-HBsAg

The Architect i1000 analyzer (Abbott Diagnostics, Abbott Park, IL, USA), which uses a chemiluminescent microparticle immunoassay (CMIA), was used to analyze HBsAg, HBeAg, and anti-HBe antibodies. Serum HBsAg was determined quantitatively, whereas serum HBeAg and anti-HBe were determined qualitatively. Analysis of these serological markers was conducted according to the manufacturer’s protocol.

### 2.3. Separation of PBMCs, Extraction and Quantitation of HBV-DNA from Plasma and from PBMCs

PBMCs were separated from whole blood using a density gradient with UNI-SEP+ tubes, according to the manufacturer’s protocol (NOVAmed Ltd., Jerusalem, Israel). To ensure the removal of residual plasma, the PBMCs cell pellet was washed with an excess of 10 mL saline. Total nucleic acids were extracted from 400 mL plasma or from PBMCs using MagDEA^®^ Dx SV kit on the magLEAD^®^ 12gC (Precision System Science Co., Ltd., Chiba, Japan), according to the manufacturer’s protocol. HBV-DNA levels of PBMCs were assessed as described [18], and the concentration was calculated in every sample per its total DNA concentration. The values are presented in copies/10^6^ cells, considering 1 μg equivalent to 150,000 cells [19].

### 2.4. HBV-RNA Quantitation

Real-time PCR (RT-PCR) was performed on both plasma and PBMCs, with extracted total nucleic acids, as previously described [14,15], using the same primers and probe with a few modifications in the reaction protocol. SensiFAST Probe Lo-ROX One-Step Kit was used (Bioline Meridian BioScience, Cincinnati, OH, USA) with suitable temperatures per the manufacturers’ instructions: 10 min at 45 °C, 2 min at 95 °C, followed by 40 cycles of 5 s at 95 °C and 20 s at 60 °C for annealing/extension in one-step PCR. In order to verify the absence of DNA contamination, all samples were amplified twice: once with the reverse transcriptase enzyme from the SensiFAST Probe Lo-ROX One-Step Kit and once without it. In order to obtain a standard RNA-based calibrator for the RT-PCR reaction, a fragment spanning the target sequence (nucleotide 1608–1933 in GenBank accession number in AB453983.1) with a T7 RNA Polymerase promoter sequence in its 5′ and a poly A tail (AAAAAAAAAAAAAAAAAAAAAAGTGAGRAGCGATAGCGTGGT) in its 3′ end was generated and cloned into a pUC57 plasmid. Using BamH1 endonuclease, the fragment was released from the plasmid and transcribed in vitro to RNA using the T7 MEGAscript kit according to the manufacturer’s instructions (Thermo Fisher, Thermo Fisher Scientific, Waltham, MA, USA). The in vitro transcribed RNA fragment was than purified, treated with RNAse free DNAse enzyme (Promega), purified again, and its concentration was determined using a Nanodrop spectrophotometer and stored at −80 °C. Limit of detection of the test was 40 copies/mL.

### 2.5. Sequence Analysis of the HBV BCP/ PC/Pol Region

When possible (HBV-DNA viral load >1000 IU/mL), amplification of HBV from plasma was performed using the following primers: for BCP and PC, 5′CTTCGCTTCACCTCTGCACGTC3′ forward primer and 5′GAGGATTAAAGACAGGTACAGTAGAAG3′ + 5′GAGGGTTAAAGACAGGTACAGTAGAAG3′ reverse primers were initially used, followed by 5′CATGGAGACCACCGTGAACGC3′ forward with 5’GTTCCCCACCTTATGAGTCC3’ reverse-nested primers. The Pol region was amplified as previously described [18]. The amplified fragments were subjected to sequencing with BigDyeDeoxy Terminators according to the manufacturer’s instructions (Applied Biosystems, Foster City, CA, USA). The resultant HBV chromatograms were cured following alignment to the reference sequence using the Open-gene system (Siemens Healthcare, Diagnostics, Deerfield, IL, USA). All curated fasta sequences were assessed by Geno2Pheno (https://hbv.geno2pheno.org/, accessed on 1 August 2021) to determine HBV genotype and escape, resistance, and BCP and PC mutations. PC mutations at positions 1814–1816 and 1874, not included in the Geno2Pheno, were assessed manually. Reference sequences used for genotyping and mutation analysis are published in the HBV Geno2Pheno website.

### 2.6. Statistics

Categorical variables were reported as numbers and percentages (%) and analyzed using Chi square of Fisher exact tests. Pearson correlation and Student’s *t*-Test were used to analyze noncategorical variables. Analyses were performed using Social Science Statistics (https://www.socscistatistics.com/tests/, accessed on 2 January 2022).

## 3. Results

### 3.1. Patients Characteristics

Overall, 139 CHB patients were recruited for the study (Table 1), most of whom were male. The majority (80.3%) originated from Israel or from Eastern Europe. Coinfection (with either HIV or Hepatitis delta) was reported for 11.5% of patients.

Most (126, 90.6%) CHB patients were HBeAg-negative (HBeAgNeg). A comparison between HBeAg-positive (HBeAgPos) and HBeAgNeg CHB patients is presented in Table 2. More HBeAgPos patients were on NA treatment (*p* < 0.05). As expected, levels of both ALT and AST liver enzymes were significantly elevated in HBeAgPos patients compared to HBeAgNeg patients (average of 63.2 and 74.8 IU/L in HBeAgPos versus 25.4 and 28.4 IU/L, respectively, *p* < 0.05). Most HBeAgPos patients had circulating viral RNA and DNA. Levels of HBV-RNA, HBV-DNA, and HBsAg (but not those of HBV-DNA in lymphocytes) were significantly higher in HBeAgPos patients compared to HBeAgNeg patients (*p* < 0.05). HBV-RNA was not detected in PBMCs.

The HBV genotype could be determined in samples from 68 patients (5 HBeAgPos and 63 HBeAgNeg). The most prevalent genotype was D (89.7%); 8.8% had genotype A. Analysis of resistance and escape mutations was possible for 63 of the patients. Only two patients had RT resistance mutations. As expected, HBV-DNA levels were high in both cases (8.2 and 6.0 log IU/ML, respectively) compared to the median HBV-DNA levels observed in the study cohort (3.5 log IU/mL). Escape mutations were found in 22.6% (13/63), with no difference in their proportions between HBeAgPos and HBeAgNeg patients. An analysis of BCP (A1762T and G1764A), PC (G1896A and G1899A), and PC initiation codon mutations (1814–1816) was performed in 62 of the samples: 4 from HBeAgPos and 58 from HBeAgNeg. The BCP and PC mutations were significantly more frequent in HBeAgNeg patients (98.3% compared to 50% in HBeAgPos, *p* < 0.01). All 57 patients with PC mutations were HBeAgNeg and had antibodies against HBeAg. Mutations at position 1874 (also related to HBeAg negativity) were not found. A list of all mutations identified is shown in Appendix A.

### 3.2. HBV Biomarkers in HBeAgNeg Patients

When HBeAgNeg patients were assessed separately, similar to the positive correlation of these markers in HBeAgPos patients, levels of HBV-RNA in samples with detectable HBV-RNA were found to positively correlate with HBV- DNA levels (r = 0.7, *p* < 0.01, Figure 1). On the other hand, no associations were identified between levels of HBV-RNA, levels of HBsAg, HBV BCP, PC, or escape mutations, and levels of HBV-DNA in PBMCs in this group of HBeAgNeg patients. However, when tested in PBMCs, the presence of HBV-DNA was significantly associated with the presence of HBV-DNA in plasma (*p* < 0.05).

Low HBsAg levels are thought to predict a future functional cure. Therefore, the results of HBeAgNeg patients were divided into those with low (log IU ≤ 3) and high (log IU > 3) HBsAg levels (Table 3). Undetectable plasma HBV-DNA and undetectable HBV-RNA, but not undetectable HBV-DNA in PBMCs, characterized patients with low levels of HBsAg (*p* < 0.05). Patients with low HBsAg were significantly older compared to those with high HBsAg (Table 3).

### 3.3. Clinical Impact of Tested Biomarkers in HBeAgNeg Patients

To better understand the potential impact of the studied biomarkers on treatment decisions, we assessed their combined results in this cohort. Patients with clear evidence of an active state of the disease (who were HBeAgPos) were not included. Of the 126 HBeAgNeg patients, 78 were currently untreated (Figure 2A). Indeed, low levels of HBsAg and undetectable plasma HBV-RNA characterized 25 of these patients. However, the other 53 patients had either high HBsAg levels (*n* = 42) or detectable plasma HBV-RNA (*n* = 11), suggesting that HBV was actively replicating in the liver. Moreover, in 13 of 25 of these patients (tested for PBMCs HBV-DNA), HBV-DNA in PBMCs was detectable.

Forty-eight HBeAgNeg patients were on NA treatment (Figure 2B). Three had high levels of HBV-DNA, HBV-RNA, and HBsAg; therefore, treatment cessation could not be potentially recommended for them. Forty-five treated patients had low HBV-DNA (<2000 IU/mL) levels. According to current guidelines, discontinuation of treatment could be recommended in non-cirrhotic HBeAgNeg patients who have achieved long-term (≥3 years) virological suppression under NA treatments if close post-NA monitoring can be guaranteed [5]. Here, thirteen of these treated patients with low HBV-DNA also had undetectable plasma HBV-RNA and low HBsAg levels (≤log3 IU/mL), and 10 of them also had undetectable HBV-DNA in lymphocytes. In all such patients, discontinuation of treatment could be discussed.

## 4. Discussion

The value of several novel non-invasive biomarkers in CHB patients has already been demonstrated in numerous studies [20]. However, currently, clinical practice is still based on the results of classical biomarkers. Moreover, an association between different non-invasive biomarkers and patient characteristics has not been rigorously demonstrated in an Israeli HBV cohort, where previous studies performed three decades ago have demonstrated the prevalence of the HBV precore mutant.

In this study, we have also shown that Israeli patients are characterized by a high prevalence of HBeAgNeg patients, and that genotype D dominates. Moreover, nearly all HBeAgNeg patients (with an HBV viral load that enabled sequencing) had BCP/PC mutations affecting HBeAg production that accounted for the loss of this viral antigen in plasma. Moreover, anti-HBe antibodies were detected in all those with PC mutations, suggesting that in the early disease stages, the original infective wildtype virus in these patients expressed HBeAg. Others have also shown that HBV genotype D prevails in countries in the Mediterranean region including Israel, and that HBeAgNeg patients are seven to nine times more frequently identified than HBeAgPos patients [17]. The PC mutation G1896A that abrogates the production of HBeAg was also shown to be highly prevalent [21,22].

In contrast with the high prevalence of BCP/PC mutations, escape mutations were found in only 13 patients. All mutations identified are located in the central core of the small HBsAg protein, between amino acids 99 and 169, a region that is exposed on the surface of the virions, and is involved in binding to anti-HBs antibodies. This is the most important antigenic determinant in envelope proteins, and point mutations in this determinant may lead to changes with regard to immunity and protection, and may result in a low affinity to neutralizing antibodies [23]. Here, the absence of association between escape mutations and other variables could imply that host factors led to their emergence rather than viral defense strategies [24].

Generally, the HBeAgPos phase was shown by others to be associated with higher levels of ALT, AST, HBV-DNA, and HBV-RNA compared to the HBeAgNeg phase. Moreover, a strong correlation between serum HBV-DNA and serum HBV-RNA levels in patients has been reported mainly in HBeAgPos patients [20]. Consistent with this, here, higher levels of HBV-RNA, HBV-DNA, and HBsAg were found in HBeAgPos patients compared to HBeAgNeg patients (*p* < 0.05). Moreover, a significant positive correlation was also found between detectable plasma HBV-DNA and HBV-RNA levels in HBeAgNeg patients. Indeed, the correlation between serum HBV-RNA and HBV-DNA was shown to be sustained after stratification into the different phases of chronic infection, although the correlation with other markers weakened [25].

As previously described, lower HBsAg levels were found in older HBeAgNeg patients [26]. Low levels of serum HBsAg (log ≤ 3 IU/mL) have been proposed to predict future HBsAg seroconversion [27]. Here, undetectable levels of plasma HBV-DNA, HBV-RNA, and HBV-DNA in PBMCs were significantly more abundant in patients with low HBsAg (≤3 log IU/mL) levels, suggesting a better outcome.

Generally, higher levels of HBV-DNA were found in plasma compared to PBMCs, and the presence of HBV-DNA in PBMCs was significantly associated with the presence of HBV-DNA in plasma. It is debatable whether the HBV detected in PBMCs reflects liver disease [28], or if HBV undergoes independent evolution in different compartments, e.g., PBMCs, under the influence of differential immune pressure [29]. As levels of HBV-DNA in PBMCs were similar in patients with both high and low HBsAg levels, independent evolution of HBV-DNA in PBMCs is hereby suggested.

The potential clinical outcome of measuring these novel biomarkers could be demonstrated in a subset of HBeAgNeg patients. It is important to establish a more accurate set of novel biomarkers as indicators for initiating antiviral treatment, specifically in this group of patients, as this condition may be associated with more active liver disease, leading to the development of cirrhosis and hepatocellular carcinoma [30]. In those who are not indicative for treatment, high HBsAg levels or the identification of circulating HBV-RNA in plasma may suggest more frequent follow-up. On the other hand, in treated HBeAgNeg patients that meet the current guidelines for stopping NA treatment, the identification of low HBsAg levels and demonstrating the absence of circulating HBV-RNA provides additional information that may assist in making such clinical decisions. Here, results of these markers may potentially affect treatment decisions in >50% (66/126) of the HBeAgNeg patients.

Absence of HBV-DNA in PBMCs may also assist when the decision to stop treatment is considered. However, the potential added clinical value of measuring HBV-DNA in PBMCs should be further assessed.

This study has several limitations. The overall size of the patient cohort was limited. However, most had the same characteristics (HBeAgNeg patients with genotype D and PC mutations), allowing us to concentrate on this group of patients, who are apparently the most common type of CHB patient in Israel [22]. In addition, the levels of hepatitis B core-related antigen (HBcrAg), a complex of three HBV proteins (HBeAg, hepatitis B core antigen, and a truncated 22 kDa precore protein), was not assessed herein. Several studies proposed measuring HBcrAg as an additional biomarker [31]. However, currently, assessment of this marker has largely been limited to Asian populations [20]. Moreover, this marker is commercially produced by a single company, its performance was recently considered debatable [32], and a new fully automated alternative was suggested; however, both commercial alternatives are not distributed in Israel.

Taken together, this study presents the added value of quantification of HBsAg and HBV-RNA on a specific HBsAgNeg cohort. Quantification of HBsAg can simply be conducted using the same technology used for qualitative HBsAg analysis [33]. Therefore, the introduction of this marker to the clinical routine should not be an obstacle. Mainly financial issues prevent its implementation in the standard clinical routine in Israel and probably also in other Western countries. HBV-RNA quantitation in plasma is indeed still challenging and should be standardized and validated on various cohorts and populations. However, it seems to be a valuable and feasible biomarker [20].

In conclusion, combined assessment of classical and novel serum biomarkers may assist when monitoring and treating CHB patients. Prospective studies are required to evaluate the clinical use of such markers for many aspects of management of CHB patients.

## Figures and Tables

**Figure 1 viruses-14-00584-f001:**
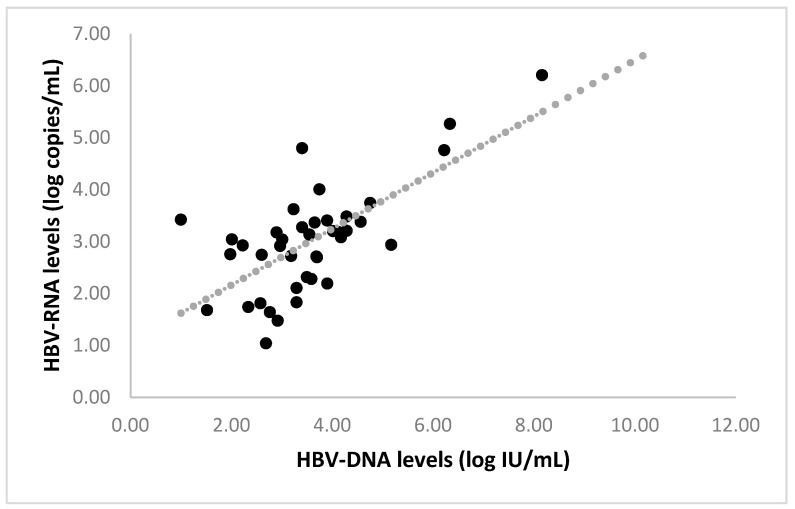
Correlation between HBV-DNA (log IU/mL) and HBV-RNA (log copies/mL) levels in HBeAgNeg patients.

**Figure 2 viruses-14-00584-f002:**
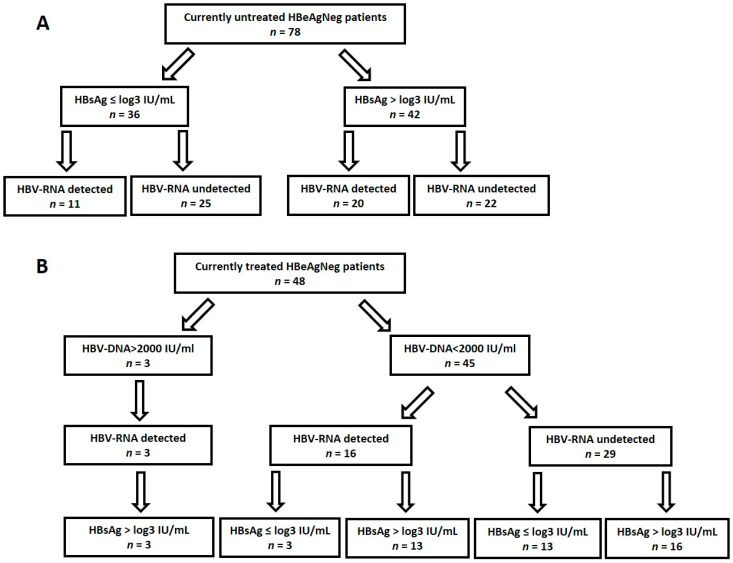
Categorization of HBsAgNeg patients by treatment, and levels of HBV-DNA, HBV-RNA and HBsAg. (**A**)—Currently untreated patients. (**B**)—Treated patients.

**Table 1 viruses-14-00584-t001:** Characterization of CHB patients (*n* = 139).

Variable		
Males, *n* (%)		94.0 (67.6)
Age, Median (IQR)		48.8 (40.1, 60.1)
Birth country, *n* (%)	Israel	52 (37.4)
	Eastern Europe	61 (43.9)
	Asia	8 (5.8)
	Africa	13 (9.4)
	Unknown	5 (3.6)
Body mass index, Median (IQR)		26.6 (23.7, 29.6)
AST, average (range)		24 (20, 30)
ALT, average (range)		25 (19, 34)
Fibrosis stage, *n* (%)	F0–F2	109 (78.4)
	F3–F4	17 (12.2)
	Unknown	13 (9.4)
Co-infection *, *n* (%)		16 (11.5)
Currently treated, *n* (%)		58 (41.7)
HBeAgPos/ HBeAgNeg		13/126 (9.4/90.6)

* Co-infection with either HDV, HIV or both.

**Table 2 viruses-14-00584-t002:** Comparison between HBeAg-positive and -negative CHB patients.

		HBeAgPos,	HBeAgNeg,	*p* Value
		*n* = 13 (9.4%)	*n* = 126 (90.6%)	
Currently treated, *n* (%)		10 (67.9)	48 (28)	<0.05
AST, average (range)		63.2 (25–250)	25.4 (9–64)	<0.05
ALT, average (range)		74.8 (24–423)	28.4 (5–115)	<0.05
Fibrosis stage, N (%)	F0–F2	9 (69.2)	100 (79.4)	0.6
	F3–F4	2 (15.4)	15 (11.9)	
	Unknown	2 (15.4)	11 (8.7)	NA
Plasma HBV-DNA	Undetectable, *n* (%)	3 (23.0)	38 (30.2)	0.6
	Detectable (*n* = 96)	4 (2.0–6.7)	3.3 (2.6, 3.8)	<0.05
	log IU/mL, Median (IQR)			
Plasma HBV-RNA	Undetectable, *n* (%)	2 (15.4)	75 (59.5)	<0.05
	Detectable (*n* = 63)	4.4 (4.1–6.6)	3.0 (2.1, 3.4)	<0.05
	log copies/mL, Median (IQR)			
PBMCs HBV-DNA	Undetectable, *n*/N (%)	5/6 (83.3)	48/74 (64.5)	0.4
	Detectable (*n* = 27)	NA	1.4 (1.2, 3.4)	NA
	log copies/10^6^ cells, Median (IQR)			
HBsAg	Log IU/mL, Median (IQR)	3.7 (3.4–4.2)	3.2 (2.3, 3.8)	<0.05
Anti-HBe Antibodies	Yes/No, *n* (%)	2/11 (15.4)	120/6 (95.2)	<0.05
Genotype (N = 68)	A, *n*/N (%)	1/5 (20.0)	5/63 (7.9)	0.4
	C, *n*/N (%)	0/5	1/63 (1.6)	NA
	D, *n*/N (%)	4/5 (80.0)	57/63 (90.4)	0.5
Mutations	BCP and PC (*n*/N, %)	2/4 (50.0)	57/58 (98.2)	<0.05
	Escape (*n*/N, %)	2/5 (40.0)	11/58 (19.0)	0.3
	Resistance (*n*/N, %)	1/5 (20.0)	1/58 (1.7)	<0.05

NA: Not applicable.

**Table 3 viruses-14-00584-t003:** HBV biomarkers in HBsAgNeg patients with low (log IU/mL ≤ 3) or high (log IU/mL > 3) HBsAg levels.

		HBsAg ≤ 3 log (*n* = 52)	HBsAg > 3 log (*n* = 74)	*p* Value
Age, average (IQR)		53.9 (41.8,61.8)	45.8 (39.0,57.5)	<0.05
ALT, average (range)		25.3 (9–116)	30.4 (5–83)	0.11
AST, average (range)		23.8 (12–48)	26.4 (9–64)	0.15
Undetectable plasma HBV-DNA, *n* (%)		33 (63.5)	20 (27.0)	<0.05
Detectable plasma HBV-DNA log, Median (IQR)		3.2 (2.6,3.8)	3.4 (2.8,3.8)	0.61
Undetectable HBV-RNA, *n* (%)		38 (73.0)	38 (51.4)	<0.05
Detectable HBV-RNA log, Median (IQR)		2.6 (1.8,3.2)	3.0 (2.3,3.4)	0.36
Undetectable PBMC HBV-DNA, *n* (%)		22/31 (70.9)	27/43 (62.8)	0.46
Detectable PBMC HBV-DNA log, Median (IQR) *		1.2 (0.9,3.5)	1.5(1.2,3.7)	0.81
Anti Hbe Antibodies		50 (96.1)	70 (94.6)	0.69
Genotype	A, *n* (%)	1/22 (4.6)	4/41 (9.8)	0.47
	C, *n* (%)	0	1/41 (2.4)	NA
	D, *n* (%)	21/22 (95.5)	36/41 (87.8)	0.32
Mutations	BCP/ PC N = 58 (n/N, %)	21/21 (100)	36/37 (98.2)	NA
	Escape, N = 58 (n/N, %)	3/20 (15)	8/38 (21.6)	0.55
	Resistance, N = 58 (n/N, %)	0/20	1/38 (2.7)	NA
Currently treated, *n* (%)		16 (30.8)	32 (43.2)	0.16

* PBMC HBV-DNA level is shown in log copies/10^6^ cells.

## Data Availability

Raw data can be provided upon request from the authors.

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
