# Peer review of "HBV-RNA, Quantitative HBsAg, Levels of HBV in Peripheral Lymphocytes and HBV Mutation Profiles in Chronic Hepatitis B"

_viruses, 2022, doi:10.3390/v14030584_

Round 1

Reviewer 1 Report

In this clinical study Yael Gozlan and colleagues investigated HBV-RNA, qHBsAg, HBV DNA in serum or plasma and in peripheral lymphocytes and HBV mutation profiles in chronic hepatitis B in Israel where most of the patients are infected with genotype D virus and mainly present with HBeAg negative CHB.  Overall, this study is well designed, executed and analyzed.

Comments:

Line 91 – “Blood was collected and serum, plasma or PBMCs were separated and stored at -20°C” – Since HBV RNA is sensitive to degradation samples are usually stored at -80°C. How soon after storage was the HBV RNA assay performed?

Methods 2.3 – lines 104-113: Was it verified somehow that HBV DNA measured from PBMCs did not originate from contamination from plasma? This has to be addressed.

Methods 2.4 – lines 114-133: Please describe the method of DNA removal from extracted total nucleic acids. Also was HBV RNA measured in PBMCs as well, as suggested in the text? If yes what were the results.

Author Response

Thank you for your review.

Line 91 – “Blood was collected and serum, plasma or PBMCs were separated and stored at -20°C” – Since HBV RNA is sensitive to degradation samples are usually stored at -80°C. How soon after storage was the HBV RNA assay performed?

Answer: Indeed, storage of any sample in -80°C is preferable, however in our experience, and especially for RT-PCR analysis, keeping in -20°C is sufficient. We usually perform the RT-PCR assay soon after extraction, and at the most within three weeks after extraction. We have similar experience both with original samples and with nucleic acids extracted from other RNA viruses like HIV and HCV HDV and others. Others also declare similar findings. For example, in a HCV paper (Raza el al 2012) plasma storage temperature at -80 and -20 °C did not affect significantly on RNA levels (p > 0.05).

Methods 2.3 – lines 104-113: Was it verified somehow that HBV DNA measured from PBMCs did not originate from contamination from plasma? This has to be addressed.

Answer: Thank you for this remark. In this study, PBMCs were separated from whole blood after removal of plasma. Then, Ficoll was added and the "white ring" formed by the PBMCs was collected and centrifuged. The resultant cell pellet was washed with 10 ml of saline, vortexed and centrifuged to remove any residual liquid. Thus, we believe that the amount of plasma was minimal and its impact on the results is minor. Indeed, we did not find significant correlation between detectable HBV-DNA levels in plasma and PBMCs. Moreover, in 4 patients HBV-DNA was detected in PBMCs and not in plasma.

To clarify the removal of plasma we added the following sentence to the methods section:

"To ensure removal or residual plasma, PBMCs cell pellet was washed with excess of 10 ml saline".

Methods 2.4 – lines 114-133: Please describe the method of DNA removal from extracted total nucleic acids. Also was HBV RNA measured in PBMCs as well, as suggested in the text? If yes what were the results.

Regarding your question on DNA removal: we did not remove DNA from the total nucleic acids. Instead, for specific detection of RNA we used RNA specific primers with polyA tail and we also compared two reactions: one with RT enzyme and the other without the RT step.  DNA removal was only done from the synthetic control, as described in the methods section.

Regarding the results of RNA in PBMCs we indeed measured HBV-RNA in PBMCs. In line 138 (results) we wrote: "HBV-RNA was not detected in PBMCs."

Reviewer 2 Report

I would add more recent reference works in the background. There are other mutations that fall and interfere with the expression of HBeAg, they have been highlighted in position 1814/1815, 1874, why have they not been researched? In sera from patients infected with these HBV variants, HBeAg is undetectable and does not even express itself on the surface of hepatocytes, which in this way are able to evade the immune response.

Author Response

I would add more recent reference works in the background. There are other mutations that fall and interfere with the expression of HBeAg, they have been highlighted in position 1814/1815, 1874, why have they not been researched? In sera from patients infected with these HBV variants, HBeAg is undetectable and does not even express itself on the surface of hepatocytes, which in this way are able to evade the immune response.

Answer to reviewer 1:

Thank you for this important remark. We have revised the introduction and added new references as suggested.

The following text was added to the introduction: Defective HBeAg secretion was mainly related of A1762T, G1764A (BCP) G1896A or G1899A (PC) mutations. PC mutations that affect the HBeAg initiation codon (positions 1814-1816) and the 1874 nonsense mutation have also been shown to lead to HBeAg sero-negativity [14, 15]. 

Regarding the other mutations suggested by you:

in the literature, most of the cases reported to be HBeAg sero-negative have the mutations that we have included in our study (A1762T, G1764A (BCP), G1896A or G1899A (PC). Regarding positions 1814/1815 and 1874 we have looked at our cases and found that 1874 site is conserved and none of our cases had mutations at this site. Most of the sequenced cases (56/62) had one or more of the mutations we assessed. Regarding the 1814/1815 and 1816 (the initiation codon mutations), we have now found that three of these six without any other mutations (A1762T, G1764A G1896A or G1899A) had one of these three initiation codon mutations. We have amended the abstract, methods, results and supplement material accordingly.

In the abstract we changed: " HBV genotype D prevailed (61/68) and >95% had BCP/PC mutations. Escape mutations were identified in 22.6% (13/63).

In the methods we added: " PC mutations at positions 1814-1816 and 1874 not included in the Geno2Pheno were assessed manually."

Results were also corrected properly.

In the supplementary material we added: " Three of the six patients without any G1896A or G1899A mutations, had mutations in the initiation codon: two with genotype D had G1816T mutation and one with genotype A had T1815C mutation."